# Causes and trends of under-five mortality in the Gilgel Gibe Health and Demographic Surveillance System, 2005 to 2019, Southwest Ethiopia: Cohort study

Desalegn Shiferaw[1,2]*, Mohammed Sanni Ali[1,3,4], Fasil Tessema[5], Bikila Regassa Feyisa[1,6], Mubarek Yesse Ashemo[1,7], Assefa Legesse Sisay[1], Muluemebet Abera[8], Tizta Tilahun Degfie[9], Esayas Alemayehu[10], Chaltu Fikru[1], Yohannes Kebede[11], Lelisa Sena Dadi[1]

1 Department of Epidemiology and Biostatistics, Jimma University, Jimma, Ethiopia, 2 Department of Public Health, Dambi Dollo University, Dambi Dollo, Ethiopia, 3 London School of Hygiene and Tropical Medicine, London, England, 4 VIBRANIUM Resources Foundation, Addis Ababa, Ethiopia, 5 CDC Ethiopia, Addis Ababa, Ethiopia, 6 School of Public Health, Wallaga University, Nekemte, Ethiopia, 7 Department of Public Health, Werabe University, Werabe, Ethiopia, 8 Department of Population and Family Health, Jimma University, Jimma, Ethiopia, 9 Department of Family Health and Population studies, Bahir Dar University, Bahir Dar, Ethiopia, 10 Faculty of Civil and Environmental Engineering, Institute of Technology, Jimma University, Jimma, Ethiopia, 11 Department of Health, Behavior, and Society, Jimma University, Jimma, Ethiopia

* latisenako@gmail.com

## Abstract

### Background

Irrespective of intensive global efforts to reduce under-five mortality, it remains a significant public health concern. Understanding the causes and trends of under-five mortality is essential for guiding targeted interventions, assessing the effectiveness of public health strategies, and monitoring changes in mortality over time. The health and demographic surveillance system is one of the preferred sources to study the cause of under-five mortality.

### Objective

This study aims to analyse the causes and trends of under-five deaths in Southwest Ethiopia using Gilgel-Gibe Health and Demographic Surveillance System (GGHDSS) database.

### Methods

GGHDSS is an open, dynamic cohort that was established in 2005. Fifteen years of mortality data and seven years of Verbal Autopsy (VA) data were extracted from the GGHDSS database for this study. The VA data are part of the fifteen year mortality dataset, in which causes of death were determined through the VA method. After

**Data availability statement:** The data used in this study are held by the Jimma University Health and Demographic Surveillance System (HDSS) Coordination Office. Data access is regulated by Jimma University and permitted through a formal data-sharing agreement. Anyone who wishes to access the data may submit a formal request to the Jimma University HDSS Coordination Office. Jimma University IRB address: Email: ero@ju.edu.et or ethicsjuirb@gmail.com website: http://www.ju.edu.et Tel. +251-4711 11457 Fax: +251471111450 P.O. Box: 378.

**Funding:** The author(s) received no specific funding for this work.

**Competing interests:** The authors have declared that no competing interests exist.

extracting the data from MYSQL and OpenHDS, it was exported to Excel for further cleaning. Finally, the cleaned data were exported to R statistical software for analysis and visualization. Neonatal, infant, and under-five mortality trends were analysed and the proportion of cause specific deaths were identified from the VA data.

## Results

Between 2005 and 2019, 28, 811 children, 13931 (48.35%) female and 14880 (51.65%) male, were born alive and registered in the GGHDSS, among which 1828 (6.34%) of them died before celebrating their fifth birthday. The overall under-five mortality rate was 63.4 (95% CIs; 60.6, 66.3) per 1000 live births while neonatal and infant mortality rates were 30(95% CIs; 28.1, 32) and 50.2 (95% CIs; 47.6, 52.7), respectively. The mortality rate during the surveillance years showed a declining trend among neonatal, infant, and overall under-five children, with average slopes of −2.49, −4.56, and −7.24, respectively. From 1070 under-five deaths captured by the VA method in GGHDSS during 2009–2016, 596 (55.7%) were male and 474 (44.30%) were female. The most common causes of neonatal death were birth asphyxia and perinatal respiratory disorders, bacterial sepsis, and prematurity including respiratory distress. In the post-neonatal period, the most common causes of death were acute lower respiratory infections (including pneumonia and acute bronchitis), intestinal infectious diseases including diarrheal diseases, and malaria. Severe malnutrition, intestinal infectious diseases, and acute lower respiratory infections were responsible for more than half of the deaths among children between 12 and 59 months of age.

## Conclusions

Under-five mortality has shown a significant declining trend between 2005 and 2019 at the study setting. Birth asphyxia, neonatal infections, and prematurity were among the most common causes of neonatal deaths, whereas infectious diseases and malnutrition were the main causes of death beyond the neonatal period. Strengthening perinatal and neonatal care, improving prevention and management of childhood infections, and enhancing early nutritional interventions are needed to reduce under-five mortality.

## Introduction

Under-five mortality refers to the death of children before celebrating their fifth birthday and is calculated per 1,000 live births [1]. Deaths among under-five children can be further disaggregated into neonatal mortality, post neonatal mortality, and child mortality [2].

Irrespective of intensive global efforts to reduce under-five mortality, it still remains a significant public health concern. In 2024, an estimated 4.9 million children died

before reaching their fifth birth day [3] The burden of under-five mortality is unevenly affecting nations where Sub-Saharan African and southern Asian countries are hardly hit [4]. The majority of these deaths were reported from Sub-Saharan Africa, a region with poor economic conditions, infrastructure, and a fragile health care system [5]. According to united Nations Inter-Agency Group for Child Mortality Estimate, more than 58% of under-five mortality in 2024 was from Sub-Saharab African region [3]. Ethiopia is one of the countries bearing the highest burden of under-five children death [1]. According to the Global Burden of Disease (GBD) 2021 report, 177,000 under-five deaths were registered in Ethiopia [5]. Furthermore, Ethiopian Demographic and Health Survey (EDHS) 2024/25 reported under-five mortality rate to be 51 deaths per1000 lve births [6]. Yet, a substantial variation in the distribution of under-five mortality was reported among the subnational of Ethiopia [7].

Sixty countries, accounting for 42% of global under five mortality, are at risk of missing the 2030 target, while 66 countries—representing 62% of under-five children and 65% of newborns—are not on track to achieve the neonatal mortality target [3]. For countries that are not on track to meet the targets, accelerated efforts are required to achieve the SDG targets on child mortality, and attaining these targets globally could avert at least 10 million under-five deaths by 2030 [8].

Children's chances of survival vary substantially by places of birth, with the highest burden of under-five deaths concentrated in Sub-Saharan Africa and South Asia [9,10]. According 2024 report, 54% of under-five mortality in Sub-Saharan Africa was due to infectious diseases while prematurity, birth asphyxia, and congenital anomalies were the top three leading cuases of death in this age group [3]. These largely preventable conditions not only result in the loss of young lives but also create significant emotional and economic strain on families, [11–13] and burden on health system especially for poor countries

Despite the local and global efforts to reduce under five mortality [10], still there are many countries far from achieving the targets [3]. Timely and reliable information on the under-five children's magnitude and causes of death plays a vital role in achieving the target of reducing deaths in this age group [14]. Like many other developing countries, Ethiopia faces a shortage of reliable data, and lacks consistent, long term datasets to assess the magnitude and to track the trend of under-five mortality [15]. Most of the available data came from small-scale surveys and health facility reports, which may not capture the true burden in rural communities.

Since recent time some higher education institution in Ethiopia have establshied Health and Demographic Surveilance System (HDSS), which serves as longitunal data source. HDSS data help bridge the gap [16] by providing a clear picture of the actual situation in the community and enabling monitoring of the trends over time [17]. Analysing HDSS data is crucial for measuring the progress toward achieving the SDG target of reducing under-five mortality to fewer than 25 deaths per 1000 live births. Therefore, this study aims to fill the gap in access to reliable evidence by leveraging HDSS data and verbal autopsy to provide robust, community-level evidence on both trends and causes of under-five mortality in Southwest Ethiopia..

## Methods

### Study setting

The study was conducted around Gilgel Gibe Hydroelectric Dam (GGHD) one, which is located 260 km to Southwest of Addis Ababa, the capital of Ethiopia, and 55 km east of Jimma City. The study kebeles (the smallest administrative structure in Ethiopia) are found in four districts surrounding the dam: Sekuru, Omo-Nada, Nadhi Gibe and Kersa. Three small towns (Dimtu, Asendabo and Deneba) and eight rural kebeles (Siba, Dogoso, Kejelo, Koticha, Ayno, Inkure, Bore and Burka) located within a 10 km radius of GGHD were included into the GGHDSS, also known as Gilgel Gibe Field Research Center (GGFRC). The enumerated baseline population of the 10 kebeles in September 2005 was 43,775 (49.53% male and 50.47% female) and the baseline of (Deneba) was taken in January 2008 and the population of the kebele was 5,993 (Fig 1).

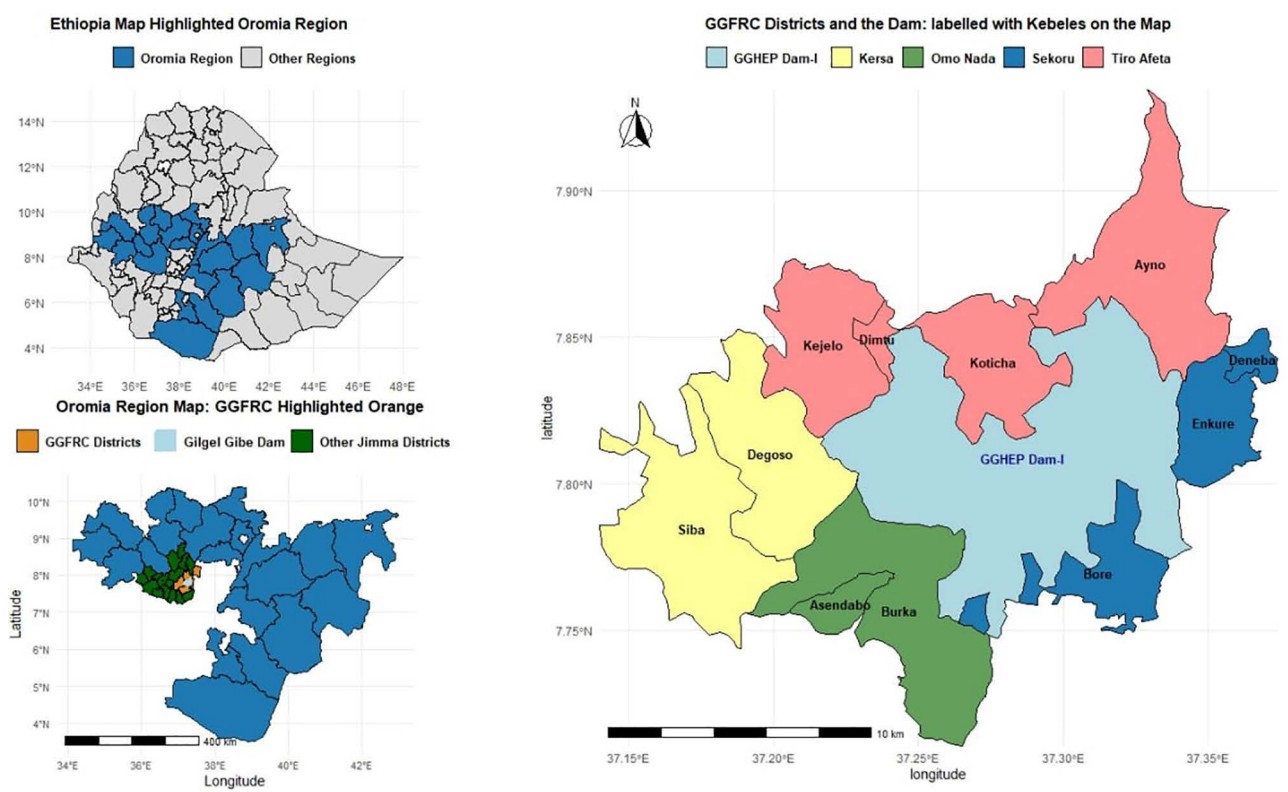

**Fig 1. Gilgel Gibe-HDSS Site Map: (Source: Gilgel Gibe-Field ResearchCenter, 2025).**

## Study design and period

The GGHDSS applies an open cohort population-based longitudinal surveillance system design. In the surveillance site, data have been collected as events occurred from 2005 to 2012, and twice a year since 2012. In this study, the trend of under-five mortality was analysed from 2005 to 2019. However, the cause of death analysis was performed for the deaths reported between 2009 and 2016 since the VA record was limited to this period.

## Study population

At the end of 2019, GGHDSS has 103,091 individuals with unique ID from which 66,279 wereactive residents of the site. The source and study population of this study were all children of under-five years, who were born to the site between 2005 and 2019. The denominator of the under-five mortality rate were all live births born to the site and joined the surveillance system by birth, and the numerator was the number of under-five deaths among those who joined the site by birth. For the cause of death analysis, deaths captured by the verbal autopsy during 2009–2016 registrations were used.

## Data collection

**Death registration.** Depending on the size of kebeles, one or two regular residents of each kebele were identified and recruited as local guides to identify, record, and report deaths that occurred in the respective kebeles using a format consisting of name, age, sex and address of the deceased individuals. Then, the local guides submit the registered deaths to the field supervisors who in turn submit the registered deaths to the research team. The report was usually

made monthly at the joint meeting of all surveillance field workers and coordinating teams. Those registered deaths were forwarded to the VA interviewers for scheduling and conducting the interviews considering forty-five days of mourning period.

**Questionnaire.** There were three different categories of VA questionnaires, namely from birth to 4-weeks of life, from4- weeks to 14-years and older than 14-years. This tool was modified and adapted from the Verbal Autopsy Interviewer's Manual, Sample Vital Registration with Verbal Autopsy [18]. The questionnaire consisted of age, sex, place of death, detailed perceived cause of death, cause of death according to the respondent, short narrative history of the circumstances before death, symptom duration, health services utilization in the period before death, and any medical evidence available at the household.

**Data collection.** The VA interviewers were at least diploma graduates who were trained on the objectives of the study: how to identify and approach respondents, how to conduct an interview, and the ethics of data collection. The training included classroom based theoretical sessions, role model interviews and field level practice. Once scheduling was done, the VA interviewers visited the deceased household and conducted the interview, supported by the local guides. Age standardized questionnaire [18], was used to collect data within one to three months after the death, considering the mourning period to minimize the distress and discomfort to respondents [19]. The trained VA data collectors interviewed a caregiver who had closely attended the deceased child, and collected information on signs and symptoms leading to the death using the adapted questionnaire.

**Assigning causes of death.** The completed questionnaires were first given to two trained physicians, who assigned probable cause of death independently and blindly. The assignment of causes of death was based on International Statistical Classification of Diseases and related health problems, 10th revision (ICD_10) [20] and based on the verbal autopsy coding scheme. After the coded causes of death based on verbal autopsy coding were compared, the cause of death was assigned based on the agreement of the two physicians' coding. In case of disagreement, those questionnaires were given to a third physician, who was still unaware of the assignment of the two physicians, for independent review. Then, the third physician coding was compared with the previous two codings; and if it agreed with either of them a specific cause of death was assigned. In case the disagreement persisted among the three physician assignments, the cause of death was considered undetermined.

**Quality assurance.** Intensive trainings were given to the physicians who reviewed the completed VA questionnaires on how to assign cause of death based on ICD-10 [20] and to the verbal autopsy interviewers. To maintain consistency, the questionnaire was translated to local languages (Afan Oromo and Amharic) and back translated to English. Furthermore, supervision and cross-checking of the filled questionnaires were done by visiting 5% of the households and re-interviewing the respondents.

## Data management and analysis

Verbal Autopsy (VA) data collected by trained interviewers were entered into a database using EpiData software, while the physician reviewed data were entered into a separate database developed for this purpose, called Y-soft. During data entry of physician reviews, identification details, background variables, and both the VA and ICD-10 codes and titles were recorded, and the data were then exported for analysis. The causes of death for each year were identified, and the proportion of deaths attributed to each cause was estimated over time. Changes in the ranking of the top five causes of under-five death were also examined.

In addition to the VA data, death data from 2005 to 2019 were extracted from the OpenHDS database. The extracted data were then checked and cleaned in Excel, then exported to R statistical software for further exploration and descriptive analysis. Although the VA data were collected on the same deceased individuals recorded in the OpenHDS data between 2009 and 2016, the two datasets were maintained separately. For this analysis, they were treated independently, and the results from both datasets were compiled to describe the cause and trends of under-five mortality.

Under-five mortality rates were calculated for different age groups: neonatal, infant, child, and the overall under-five mortality. The mortality rate for each age group was computed by dividing the number of deaths in that age group for a given year by the total number of live births in the same year. Furthermore, Mann–Kendall Trend Test was used to check whether the observed declining pattern was significant or not. Trends in under-five mortality were visualised using line graphs.

### Ethical considerations

The data we used in this study were from Gilgel Gibe Health and Demographic Surveillance System database collected for fifteen years, which is administered by Jimma University. All data provided to the investigators were fully anonymized prior to access, and no direct or indirect personal identifiers were available to the research team at any stage of analysis. As a result, individual participants couldn't be identified. The study was carried out after obtaining approval for the use of data from GGHDSS coordination office by submitting a data use agreement.

The analysis was conducted after getting approval from the Institutional Review Board (IRB) of Jimma University Institute of Health with protocol number of JUIH/IRB/0428/24. The data for this study was accessed on 05/02/2025.

## Results

### Socio-demographic characteristics of the deceased, GGHDSS, Ethiopia

During the fifteen years of surveillance period, 2005–2019, there were 28,811 live births and 1,828 (6.34%) under-five children's deaths among those born to the site. This accounted for about 34.64% of the total deaths recorded (5,277). The VA data recorded for under-five deaths during 2009–2016 were 1299 deaths, from which 229 of them were stillbirths. Therefore, 1070 of the VA data were used for cause of death analysis. From the total 1070 under-five deaths captured by the VA method in GGHDSS during 2009–2016, 596 (55.7%) were male and 474 (44.30%) were female. From the VA data, 165(15.42%) were from urban areas while the remaining 905(84.60%) were from rural kebeles. Furthermore, 492 (45.98%) were in their first month of life, 302(28.22%) were in post-neonatal age, and 276 (25.79%) were between one and four-years old. Among the neonatal deaths, 256 (52.03%) were very early neonatal deaths (occurred within the first twenty-four hours) and 125 (25.41%) were early neonatal deaths which happened within the first week of life. Furthermore, 390 (36.45%) of the deceased had visited the health facility, while 645 (60.30%) did not visit; the status of the remaining 35 (3.3%) was not known. The majority of deaths, 908 (84.9%) occurred at home while only 130 (12.15%) deaths occurred at a health facility. On the other hand, respondents did not know the place of death for 20 deaths (1.87%) while the data for the remaining 12 deaths were missing (Table 1).

### Causes of under-five children's death using VA data

Among the neonatal age group, birth asphyxia and perinatal respiratory disorders accounted for 182 (37%), while 156 (31.70%) of the deaths were caused by bacterial sepsis of the newborn. Prematurity including respiratory distress accounted for 103 (20.93%) deaths.

In the post-neonatal period, 92 (30.50%) of the deaths were due to acute lower respiratory infections including pneumonia and acute bronchitis. Intestinal infectious diseases including diarrheal diseases and malaria accounted for 66 (21.90%) and 44 (14.60%) of the deaths, respectively. Among children aged from 12 to 59 months, 52 (18.80%) deaths were caused by severe malnutrition, 47 (17.00%) were caused by intestinal infectious diseases including diarrheal diseases, and 43 (15.60%) were caused by acute lower respiratory infections including pneumonia and acute bronchitis (Table 2).

The following figure is a stacked bar graph showing the top five causes of under-five deaths in GG-HDSS, Southwest Ethiopia. Birth asphyxia was the leading cause of death, especially among the neonatal age group (Fig 2).

Table 1. Socio-demographic characteristics of the deceased in GGHDSS, verbal autopsy, 2009 to 2016.

| Characteristics | Category | | Age group | | | |
| --- | --- | --- | --- | --- | --- | --- |
| | | Neonatal | Post-neonatal | Infant | Child | Overall under-five children |
| | | Frequency (%) | Frequency (%) | | Frequency (%) | Frequency (%) |
| Sex | Male | 297 (60.4) | 165 (54.6) | 462 (58.2) | 134 (48.6) | 596(55.7) |
| | Female | 195 (39.6) | 137 (45.4) | 332(41.8) | 142 (51.4) | 474(44.3) |
| Residence | Urban | 74 (15.0) | 47 (15.6) | 121(15.2) | 44 (15.9) | 165(15.4) |
| | Rural | 418 (85.0) | 255 (84.4) | 673(84.8) | 232 (84.1) | 905(84.6) |
| Visited health facility | Yes | 49 (10.0) | 164 (54.3) | 213(26.8) | 177(64.1) | 390(36.4) |
| | No | 415(84.3) | 135(44.7) | 550(69.3) | 95(34.4) | 645(60.3) |
| | Not known | 28(5.7) | 3 (1.0) | 31(3.9) | 4 (1.5) | 35(3.3) |
| Place of death | Home | 413(83.9) | 260(86.1) | 673(84.8) | 235(85.1) | 908(84.9) |
| | Health facility | 71(14.4) | 34(11.3) | 105(13.2) | 25(9.1) | 130(12.1) |
| | Don't know | 1(0.2) | 6(2.0) | 7(0.9) | 13(4.7) | 20(1.9) |
| | Not available | 7(1.4) | 2(0.7) | 9(1.1) | 3(1.1) | 12(1.1) |

The following figure shows the trend of each of the top five causes of under-five deaths over the seven year period (2009–2016). As depicted in the graph, the causes showed a brief time of an increasing trend till 2011. The number of death caused by birth asphyxia and perinatal respiratory disorder reached its peak in 2013 (Fig 3).

### Trend of under-five mortality rate, 2005–2019

The overall under-five mortality rates have shown a decreasing trend during the follow up time. However, the decreasing trend was not uniform; rather, there was fluctuation in the trends of the mortality rate. The overall under-five, infant, and neonatal mortality rates were 63.4 with 95%CIs (60.6, 66.3), 50.2 (47.6, 52.7) and 30 (28.1, 32) per 1000 live births, respectively. The mortality rate was highest during 2008 in all age categories. The slopes of mortality reduction were −2.49, −4.56, and −7.24 among neonatal, infant, and under-five children, respectively (Table 3).

The following graph visualises the overall decreasing trend of mortality rate among the three age categories, namely neonatal mortality, infant mortality, and overall under-five mortality, from 2005 to 2019. The decreasing trends in the three age groups are: neonatal (tau value = −0.842, P < 0.001, infant (tau value = −0.829, P < 0.001 and the under-five (tau value = −0.905, P < 0.001) and they showed a significant decline (**Fig 4**).

The under-five mortality rate exhibited slight fluctuations over the study period. As shown in Fig 5, the observed mortality rate demonstrated year to year variability especially till 2013. To better understand the underlying trend, a LOESS (Locally Estimated Scatterplot Smoothening) model was applied, which revealed a relatively smooth declining pattern. The smoothed trend reduces the impact of short term fluctuations and confirms that, despite some irregularities in the observed data, under-five mortality has significantly decreased over the 15 years period (**Fig 5**).

### Discussion

This study investigated the trend of under-five mortality during 2005–2019 and the causes of death using verbal autopsy during 2009–2016. The finding of this study shows the pre COVID-19 causes and trends of under-five mortality, which may serve as a baseline for studies evaluating the impact of the pandemic on under-five mortality in Ethiopia. Accordingly, during the 2005–2019 period, the infant and neonatal mortality rates were 50.2 and 30 deaths per 1000 live births, respectively. The highest mortality rates were reported in 2005 in all age groups. The overall top five causes of death were birth asphyxia and perinatal respiratory disorders, bacterial sepsis of newborn, prematurity including respiratory distress, acute

**Table 2. Distribution of causes of death by age in GG-HDSS, Southwest Ethiopia, 2009- 2016.**

| Causes of death | Frequency (%) |
|---|---|
| | |
| **Neonatal (n = 492)** | |
| Birth asphyxia and perinatal respiratory disorders | 182 (37.0) |
| Bacterial sepsis of new born | 156 (31.7) |
| Prematurity (including respiratory distress) | 103 (20.9) |
| Undetermined | 14 (2.8) |
| Other diseases related to the perinatal period, unspecified | 14 (2.8) |
| Other | 23 (4.7) |
| **Post-neonatal (n = 302)** | |
| Acute lower respiratory infections (including pneumonia and acute bronchitis) | 92 (30.5) |
| Intestinal infectious diseases (including diarrheal diseases) | 66 (21.9) |
| Malaria | 44 (14.6) |
| Tuberculosis | 22 (7.3) |
| Undetermined | 19 (6.3) |
| Unspecified cause of death | 16 (5.3) |
| Severe malnutrition | 7 (2.3) |
| Measles | 6 (2.0) |
| Nutritional anemia | 6 (2.0) |
| Bacterial sepsis of new-born | 5 (1.7) |
| Other | 19 (6.3) |
| **Child (n = 276)** | |
| Severe malnutrition | 52 (18.8) |
| Intestinal infectious diseases (including diarrheal diseases) | 47 (17.0) |
| Acute lower respiratory infections (including pneumonia and acute bronchitis) | 43 (15.6) |
| Malaria | 38 (13.8) |
| Nutritional anemia | 27 (9.8) |
| Tuberculosis | 18 (6.5) |
| Measles | 8 (2.9) |
| Accidental drowning and submersion | 7 (2.5) |
| Meningitis | 5 (1.8) |
| Undetermined | 5 (1.8) |
| Other † | 26 (9.4) |

† the 'other' category contains Unspecified cause of death, asthma, Epilepsy, acute abdomen, pedestrian injured in traffic accident,respiratory failure, not elsewhere classified, tetanus (excluding tetanus neonatorum),accident, unspecified, accidental poisoning and exposure to noxious substance, congenital malformations of the nervous system, and contact with venomous animals and plants.

lower respiratory infections including pneumonia and acute bronchitis, and intestinal infectious diseases including diarrheal diseases.

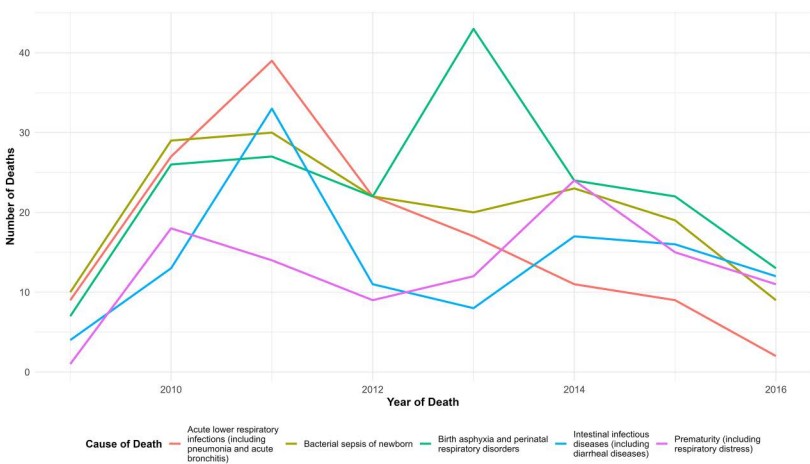

**Fig 2. Top five causes of death among under-five children, GGHDSS, 2009 to 2016.**

**Fig 3. Trend of the top five-causes of under-five death, GGHDSS, 2009 to 2016.**

**Table 3. Trend of under-five mortality in GG-HDSS, Southwest Ethiopia, 2005-2019.**

| Year | Live Births | Neonatal | | Infant | | Under-five | |
|---|---|---|---|---|---|---|---|
| | | Deaths | NMR (95%CIs) | Deaths | MR (95%CIs) | Deaths | MR (95%CIs) |
| 2005 | 443 | 26 | 58.7 (36.8, 80.6) | 40 | 90.3 (63.6, 117) | 57 | 128.7 (97.5, 159.8) |
| 2006 | 1612 | 64 | 39.7(30.2, 49.2) | 95 | 58.9(47.4, 70.4) | 150 | 93.1 (78.9, 107.2) |
| 2007 | 1935 | 85 | 43.9(34.8, 53.1) | 152 | 78.6(66.6, 90.5) | 205 | 105.9(92.2, 119.7) |
| 2008 | 1915 | 94 | 49.1(39.4, 58.8) | 162 | 84.6(72.1, 97.1) | 198 | 103.4(89.8, 117) |
| 2009 | 1877 | 64 | 34.1(25.9, 42.3) | 109 | 58.1(47.5, 68.7) | 137 | 73(61.2, 84.8) |
| 2010 | 2169 | 71 | 32.7(25.2, 40.2) | 138 | 63.6(53.4, 73.9) | 178 | 82.1(70.5, 93.6) |
| 2011 | 1926 | 63 | 32.7(24.8, 40.7) | 120 | 62.3(51.5, 73.1) | 148 | 76.8(64.9, 88.7) |
| 2012 | 1865 | 56 | 30(22.3, 37.8) | 88 | 47.2(37.6, 56.8) | 108 | 57.9(47.3, 68.5) |
| 2013 | 2243 | 81 | 36.1(28.4, 43.8) | 129 | 57.5(47.9, 67.1) | 157 | 70(59.4, 80.6) |
| 2014 | 2143 | 69 | 32.2(24.7, 39.7) | 104 | 48.5(39.4, 57.1) | 120 | 56(46.3, 65.7) |
| 2015 | 2267 | 61 | 26.9(20.2, 33.6) | 97 | 42.8(34.5, 51.1) | 122 | 53.8(44.5, 63.1) |
| 2016 | 2379 | 56 | 23.5(17.4, 29.6) | 92 | 38.7(30.9, 46.4) | 114 | 47.9(39.3, 56.5) |
| 2017 | 1890 | 27 | 14.3(8.9, 19.6) | 41 | 21.7(15.1, 28.3) | 49 | 25.9(18.8, 33.1) |
| 2018 | 2246 | 30 | 13.4(8.6, 18.1) | 48 | 21.4(15.4, 27.4) | 55 | 24.5(18.1, 30.9) |
| 2019 | 1901 | 18 | 9.5(5.1, 13.8) | 30 | 15.8(10.2, 21.4) | 30 | 15.8(10.2, 21.4) |
| Total | 28811 | 865 | 30 (28.1, 32) | 1,445 | 50.2(47.6, 52.7) | 1,828 | 63.4(60.6, 66.3) |
| Average slope | | −2.49 | MR with 95%CIs | −4.58 | MR with 95%CIs | −7.24 | MR with 95%CIs |

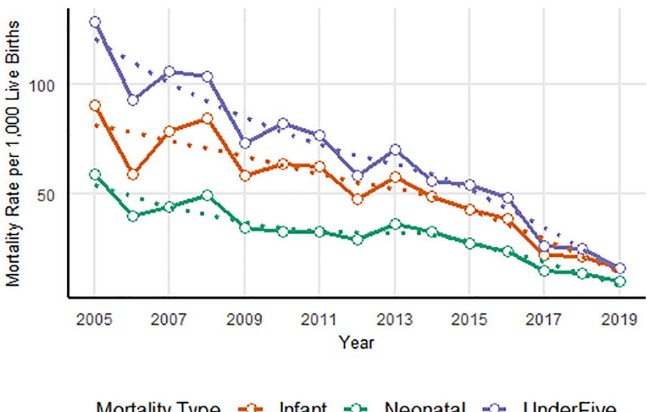

**Fig 4. Trends in Neonatal, Infant, and under-five Mortality Rates in GGHDSS, Southwest Ethiopia; 2005-2019.**

The under-five mortality rate in this study, 63.4 per 1000 live births, was higher than the findings of Global burden of disease 2019 [7], 52.4 per 1000 live births, and other studies conducted in northern Ethiopia, 35.6 128.7 per 1000 live births [21,22], eastern Ethiopia, 46.3 per 1000 live births [23], and mini EDHS, 59 deaths per 1000 live births [24]. On the other hand the findings of this study were lower compared to the magnitude of under-five mortality rate reported in the Ethiopian Demographic and Health Survey (EDHS) 2016, which was 67 per 1000 live births [25].

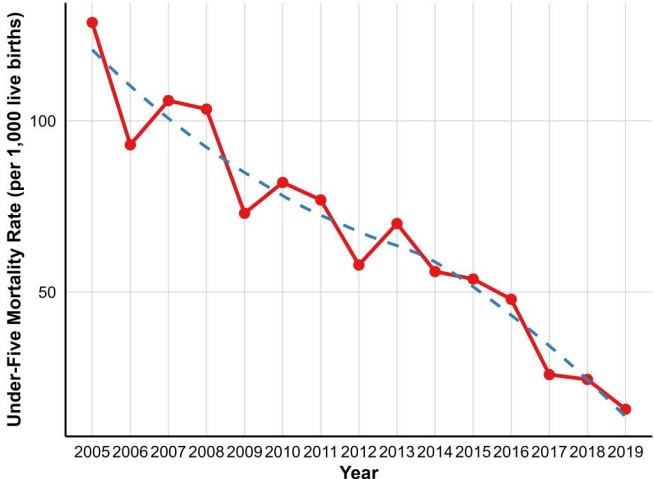

**Fig 5. Comparison of Observed and LOESS predicted Under-five Mortality Rates in GGHDSS, Southwest Ethiopia, 2005-2019.**

The possible reasons for this difference might be due to differences in calendar time as under-five mortality rate has declined over time. It could also be due to the spatial difference in under-five mortality between different regions of the country [26].

The majority of the deaths recorded through the VA occurred in rural areas. This finding was in line with the findings of the study in Tigray region Kilite Awlaelo HDSS where more than 97% of the deaths recorded were from rural areas [21]. The possible reason why rural residents are affected might be from the nature of HDSS, which is predominantly composed of the rural population. For example, more than 85% of the population in the current study are rural residents. Furthermore, these surveillance sites primarily targeted the rural kebeles unlike other HDSS sites which have lately included urban kebeles [17]. On the other hand, the level of access to health facilities is comparatively low in rural kebeles, infrastructures and transportation challenges might have led to either late or lower healthcare -seeking in where the children might have died from curable health problems.

This study revealed that more than 45% of the total deaths were among the neonatal age group of which more than half deaths occurred within the first twenty four hours of life. This finding was similar with that of the study conducted in Tigray Region using the HDSS data which reported that 46.73% of the deaths occurred among neonatal age group [21], and a report by United Nations Inter-Agency Group for Child Mortality Estimation where nearly about 48% of the total under-five mortality was among neonatal age group [1]. This high proportion of neonatal deaths from the total under-five deaths is the true reflection of the persistently higher proportion of deaths of neonates compared with the other age groups [24].

Regarding the place of death, 84.9% of the total deaths occurred at home. This finding is consistent with a study conducted in Eastern Ethiopia, where the majority of the neonatal deaths occurred at home, 94% [27]. The similarity between the two studies could be attributed to the fact that both studies utilised HDSS data, which often reflect rural and semi-urban populations with limited access to health care facilities. However, the proportion of home deaths in the present study is notably higher than that reported in a longitudinal study from South Africa between 2000 and 2015, where only 53% of deaths occurred at home [28] and that of the community based study conducted in Northern Ethiopia, 61.6% [22]. The difference with the findings of the South Africa study might be explained by the differences in the strength of the healthcare systems and levels of economic development between the two countries. Regarding the study conducted in

Northern Ethiopia, there are still regional variations in healthcare accessibility and community awareness across different parts of the country. Additionally, this may be due to poor accessibility of health care services in the surveillance area, whereas the study conducted in Northern Ethiopia included the entire region of Tigray.

The other alarming finding in this study was that 60.28% of the deceased under-five children and 84.20% of neonates did not visit health facilities. This was far from the finding of a study conducted in rural South Africa, where 84% of those children who died at home had accessed health care during their final illnesses [28]. The observed discrepancy between the findings of the two studies may be attributed to several contextual and systemic factors such as differences in health system infrastructure and accessibility, socio-cultural beliefs and perceptions of illness, awareness and health literacy levels, and economic and logistic barriers. A study conducted in Ethiopia reported that low levels of awareness, low socio-economic status, geographical inaccessibility, barriers related to a deficient healthcare system, community perceptions, and cultural restrictions as the major obstacles to healthcare utilization for under-five children [29].

This study found that the leading causes of under-five mortality were birth asphyxia and perinatal respiratory disorders, bacterial sepsis of newborn, prematurity, acute lower respiratory infections, and intestinal infectious diseases. These causes of death overlap with the finding of the studies conducted in different parts of Ethiopia [21,22,30–33]. This overlap suggests that under-five mortality in different parts of Ethiopia may share common underlying causes, reflecting broader structural and systematic health challenges. These may include inadequate maternal and neonatal care, poor infection prevention practices, and limited access to health services, and undernutrition. The proportion of undetermined cause of death reported in this study is consistent with the findings from other HDSS sites [27,34]. According to ICD-10 codes, cause of death is labelled as undetermined if disagreement persisted among the three physician assignments [18,20].

The high burden of neonatal deaths highlights the need to strengthen perinatal and neonatal care, including skilled delivery and early post neonatal services. The continued contribution of infectious diseases among post neonatal and child age groups calls for integrated interventions, such as improved interventions, such as improved immunization, nutrition programs, WASH, and malaria prevention. The finding that over 60% of deceased children didn't visit a health facility underscores significant barriers to access, requiring policies that improve service availability, affordability, and community level care.

As verbal autopsy data were only available for part of the surveillance period, the surveillance site should regain the verbal autopsy data collection to ensure more complete and continuous cause of death data. Further research is also needed to explore the underlying reasons for low health facility utilization, including access, socio-economic, behavioral factors.

### Strengths and Limitations of the study

The strength of this study was that it used more than fifteen years of surveillance data, which has been collected twice a year and able to catch vital events for years. Verbal autopsy gives a good opportunity to follow the trend of the causes of under five deaths, which provides the policy makers reliable evidence to act accordingly. One of the limitations of this study was that the VA data are not available for the entire surveillance period and the VA related analysis was limited to 2009–2016. This may introduce temporal selection bias in which the cause of death identified through VA might not be generalizable to the period where data is missing.

## Conclusions

Neonatal deaths accounted for more than 45% of the total death, with a substantial proportion occurring within the first 24 hours of life. Birth asphyxia, neonatal infections, and prematurity dominated neonatal deaths, whereas infectious diseases and malnutrition were the main causes of death beyond the neonatal period. This study revealed a declining trend in under-five mortality between 2005 and 2019. Strengthening perinatal and neonatal care, improving prevention and management of childhood infections, and enhancing early nutritional interventions can significantly reduce under-five mortality.

## Acknowledgments

We are grateful to the GGHDSS community for their valuable information during the surveillance period. We also extend our sincere thanks to all those who participated in data collection, supervision, and management. The support of Lami Diriba during the process of data extraction was invaluable. Finally, we thank the GGHDSS coordination office for supporting us in accessing the database.

## Author contributions

**Conceptualization:** Desalegn Shiferaw, Mohammed Sanni Ali, Bikila Regassa Feyisa, Lelisa Sena Dadi.

**Data curation:** Desalegn Shiferaw, Bikila Regassa Feyisa, Mubarek Yesse Ashemo, Assefa Legese Sisay.

**Formal analysis:** Desalegn Shiferaw.

**Investigation:** Desalegn Shiferaw, Lelisa Sena Dadi.

**Methodology:** Desalegn Shiferaw, Tizta Tilahun Degfie, Lelisa Sena Dadi.

**Project administration:** Desalegn Shiferaw, Mohammed Sanni Ali, Fasil Tessema, Lelisa Sena Dadi, Chaltu Fikru.

**Software:** Desalegn Shiferaw, Bikila Regassa Feyisa, Mubarek Yesse Ashemo, Assefa Legese Sisay.

**Supervision:** Mohammed Sanni Ali, Fasil Tessema, Muluemebet Abera, Tizta Tilahun Degfie, Esayas Alemayehu, Yohannes Kebede, Lelisa Sena Dadi, Chaltu Fikru.

**Validation:** Desalegn Shiferaw, Mohammed Sanni Ali, Lelisa Sena Dadi.

**Visualization:** Desalegn Shiferaw, Lelisa Sena Dadi.

**Writing – original draft:** Desalegn Shiferaw, Lelisa Sena Dadi.

**Writing – review & editing:** Desalegn Shiferaw, Mohammed Sanni Ali, Fasil Tessema, Bikila Regassa Feyisa, Mubarek Yesse Ashemo, Assefa Legese Sisay, Muluemebet Abera, Tizta Tilahun Degfie, Esayas Alemayehu, Yohannes Kebede, Lelisa Sena Dadi, Chaltu Fikru.

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
