## [Decision Letter · Decision Letter 0]

5 Nov 2025

PONE-D-25-49989Causes and Trends of Under-five Mortality in Gilgel Gibe Health and Demographic Surveillance System, Southwest EthiopiaPLOS ONE

Dear Dr. Shiferaw,

Thank you for submitting your manuscript to PLOS ONE. After careful consideration, we feel that it has merit but does not fully meet PLOS ONE’s publication criteria as it currently stands. Therefore, we invite you to submit a revised version of the manuscript that addresses the points raised during the review process.

**Please address the reviewer comments and revise your references to include the most recent sources where possible. Additionally, kindly correct any typographical and grammatical errors throughout the document.**

We look forward to receiving your revised manuscript.

Kind regards,

Addis Eyeberu

Academic Editor

PLOS ONE

**Journal Requirements:**

1. When submitting your revision, we need you to address these additional requirements. Please ensure that your manuscript meets PLOS ONE's style requirements, including those for file naming. The PLOS ONE style templates can be found at https://journals.plos.org/plosone/s/file?id=wjVg/PLOSOne_formatting_sample_main_body.pdf and https://journals.plos.org/plosone/s/file?id=ba62/PLOSOne_formatting_sample_title_authors_affiliations.pdf 2. Please provide additional details regarding participant consent. In the ethics statement in the Methods and online submission information, please ensure that you have specified (a) whether consent was informed and (b) what type you obtained (for instance, written or verbal, and if verbal, how it was documented and witnessed). If your study included minors, state whether you obtained consent from parents or guardians. If the need for consent was waived by the ethics committee, please include this information. If you are reporting a retrospective study of medical records or archived samples, please ensure that you have discussed whether all data were fully anonymized before you accessed them and/or whether the IRB or ethics committee waived the requirement for informed consent. If patients provided informed written consent to have data from their medical records used in research, please include this information Once you have amended this/these statement(s) in the Methods section of the manuscript, please add the same text to the “Ethics Statement” field of the submission form (via “Edit Submission”). For additional information about PLOS ONE ethical requirements for human subjects research, please refer to http://journals.plos.org/plosone/s/submission-guidelines#loc-human-subjects-research. 3. We note that you have indicated that there are restrictions to data sharing for this study. For studies involving human research participant data or other sensitive data, we encourage authors to share de-identified or anonymized data. However, when data cannot be publicly shared for ethical reasons, we allow authors to make their data sets available upon request. For information on unacceptable data access restrictions, please see http://journals.plos.org/plosone/s/data-availability#loc-unacceptable-data-access-restrictions.  Before we proceed with your manuscript, please address the following prompts: a) If there are ethical or legal restrictions on sharing a de-identified data set, please explain them in detail (e.g., data contain potentially identifying or sensitive patient information, data are owned by a third-party organization, etc.) and who has imposed them (e.g., a Research Ethics Committee or Institutional Review Board, etc.). Please also provide contact information for a data access committee, ethics committee, or other institutional body to which data requests may be sent. b) If there are no restrictions, please upload the minimal anonymized data set necessary to replicate your study findings to a stable, public repository and provide us with the relevant URLs, DOIs, or accession numbers. Please see http://www.bmj.com/content/340/bmj.c181.long for guidelines on how to de-identify and prepare clinical data for publication. For a list of recommended repositories, please see https://journals.plos.org/plosone/s/recommended-repositories. You also have the option of uploading the data as Supporting Information files, but we would recommend depositing data directly to a data repository if possible. Please update your Data Availability statement in the submission form accordingly. 4. We note that Figure 1 in your submission contain map images which may be copyrighted. All PLOS content is published under the Creative Commons Attribution License (CC BY 4.0), which means that the manuscript, images, and Supporting Information files will be freely available online, and any third party is permitted to access, download, copy, distribute, and use these materials in any way, even commercially, with proper attribution. For these reasons, we cannot publish previously copyrighted maps or satellite images created using proprietary data, such as Google software (Google Maps, Street View, and Earth). For more information, see our copyright guidelines: http://journals.plos.org/plosone/s/licenses-and-copyright. We require you to either present written permission from the copyright holder to publish these figures specifically under the CC BY 4.0 license, or remove the figures from your submission: a. You may seek permission from the original copyright holder of Figure 1 to publish the content specifically under the CC BY 4.0 license.   We recommend that you contact the original copyright holder with the Content Permission Form (http://journals.plos.org/plosone/s/file?id=7c09/content-permission-form.pdf) and the following text:“I request permission for the open-access journal PLOS ONE to publish XXX under the Creative Commons Attribution License (CCAL) CC BY 4.0 (http://creativecommons.org/licenses/by/4.0/). Please be aware that this license allows unrestricted use and distribution, even commercially, by third parties. Please reply and provide explicit written permission to publish XXX under a CC BY license and complete the attached form.” Please upload the completed Content Permission Form or other proof of granted permissions as an "Other" file with your submission. In the figure caption of the copyrighted figure, please include the following text: “Reprinted from [ref] under a CC BY license, with permission from [name of publisher], original copyright [original copyright year].” b. If you are unable to obtain permission from the original copyright holder to publish these figures under the CC BY 4.0 license or if the copyright holder’s requirements are incompatible with the CC BY 4.0 license, please either i) remove the figure or ii) supply a replacement figure that complies with the CC BY 4.0 license. Please check copyright information on all replacement figures and update the figure caption with source information. If applicable, please specify in the figure caption text when a figure is similar but not identical to the original image and is therefore for illustrative purposes only.The following resources for replacing copyrighted map figures may be helpful: USGS National Map Viewer (public domain): http://viewer.nationalmap.gov/viewer/The Gateway to Astronaut Photography of Earth (public domain): http://eol.jsc.nasa.gov/sseop/clickmap/Maps at the CIA (public domain): https://www.cia.gov/library/publications/the-world-factbook/index.html and https://www.cia.gov/library/publications/cia-maps-publications/index.htmlNASA Earth Observatory (public domain): http://earthobservatory.nasa.gov/Landsat:
http://landsat.visibleearth.nasa.gov/USGS EROS (Earth Resources Observatory and Science (EROS) Center) (public domain): http://eros.usgs.gov/#Natural Earth (public domain): http://www.naturalearthdata.com/ 5. If the reviewer comments include a recommendation to cite specific previously published works, please review and evaluate these publications to determine whether they are relevant and should be cited. There is no requirement to cite these works unless the editor has indicated otherwise.

**Additional Editor Comments:**

Please address the reviewer comments and revise your references to include the most recent sources where possible. Additionally, kindly correct any typographical and grammatical errors throughout the document.

Reviewers' comments:

Reviewer's Responses to Questions

**Comments to the Author**

1. Is the manuscript technically sound, and do the data support the conclusions?

Reviewer #1: Yes

Reviewer #2: Yes

2. Has the statistical analysis been performed appropriately and rigorously? 

Reviewer #1: Yes

Reviewer #2: Yes

3. Have the authors made all data underlying the findings in their manuscript fully available?

Reviewer #1: Yes

Reviewer #2: No

4. Is the manuscript presented in an intelligible fashion and written in standard English?

Reviewer #1: Yes

Reviewer #2: Yes

5. Review Comments to the Author

**Reviewer #1:** Thank you for efforts towards ending under five mortality in Ethiopia.

This manuscript should be editted by professionals to improve its grammar.

Clearly highlight the type of study this is, in the title whether cohort, case control or crossectional study.

Also avoid using acronyms before stating them in full as the case in lines 79 and 88

Kindly state the bias arising out of the stated limitations

**Reviewer #2:** The title should be adjusted to: Causes and Trends of Under-five Mortality in Gilgel Gibe Health and Demographic

Surveillance System 2005 to 2019, Southwest Ethiopia. The rest of the article is good and scientifically sound.

6. PLOS authors have the option to publish the peer review history of their article (what does this mean?). If published, this will include your full peer review and any attached files.

Reviewer #1: No

Reviewer #2: **Yes:** Dr Baboucarr Cham

---

## [Author Response · Author response to Decision Letter 1]

5 Jan 2026

Reviewers' comments:

Reviewer #1: Thank you for efforts towards ending under five mortality in Ethiopia.

This manuscript should be editted by professionals to improve its grammar.

Response: Thank you for your comments. The manuscript has been edited by a professional editor, and the grammar and language have now been improved

Clearly highlight the type of study this is, in the title whether cohort, case control or crossectional study.

Response: thank you for your invaluable suggestion. The title is modified to ‘Causes and Trends of Under-five Mortality: A Retrospective Cohort Study Using Gilgel Gibe Health and Demographic Surveillance System 2005 to 2019, Southwest Ethiopia’.

Also avoid using acronyms before stating them in full as the case in lines 79 and 88

Response: Thank you for your comments. All the comments have been thoroughly addressed in the revised manuscript.

Kindly state the bias arising out of the stated limitations

Response: thank you very much for your critical comment and it is corrected accordingly. The following statement is included in the manuscript to state a bias that may arise: ‘This may introduce temporal selection bias in which the cause of death identified through VA might not be generalizable to the period where data is missing’

Reviewer #2: The title should be adjusted to: Causes and Trends of Under-five Mortality in Gilgel Gibe Health and Demographic

Surveillance System 2005 to 2019, Southwest Ethiopia. The rest of the article is good and scientifically sound.

Response: Thank you very much for your insightful comment. The recommendation has been considered and incorporated accordingly.

---

## [Decision Letter · Decision Letter 1]

24 Mar 2026

PONE-D-25-49989R1Causes and Trends of Under-five Mortality: A Retrospective Cohort Study Using Gilgel Gibe Health and Demographic Surveillance System, 2005 to 2019, Southwest EthiopiaPLOS One

Dear Dr. Shiferaw,

Thank you for submitting your manuscript to PLOS ONE. After careful consideration, we feel that it has merit but does not fully meet PLOS ONE’s publication criteria as it currently stands. Therefore, we invite you to submit a revised version of the manuscript that addresses the points raised during the review process.

Please criticaly adress the following comments. 

Dear authors, thank you for your revisions. Please address the following points to make your article rigorous and publishable.

**Modify the title for clarity and consistency.**

The title currently reads: *"Causes and Trends of Under-five Mortality: A Retrospective Cohort Study Using Gilgel Gibe Health and Demographic Surveillance System, 2005 to 2019, Southwest Ethiopia"*.

I suggest revising it to a more concise and standard format: *"Causes and Trends of Under-five Mortality in the Gilgel Gibe Health and Demographic Surveillance System, Southwest Ethiopia: Cohort Study"*. This places the study design at the end, which aligns with PLOS ONE style.

**2. Strengthen the introduction to better articulate the problem, policy efforts, gaps, and study contribution.**

The introduction should be restructured to clearly guide the reader through:

Burden of the problem: Global and national under-five mortality figures, with emphasis on Ethiopia's context and regional disparities.

Causes and consequences: Briefly highlight the leading causes of death and their impact on families, health systems, and national development.

Policy efforts and gaps: Discuss existing national and global strategies (e.g., SDGs, Child Survival Action), and acknowledge that despite these efforts, progress remains uneven. Emphasize the persistent data gaps—particularly the lack of reliable, long-term, community-level data on causes and trends.

What has not been done: Note that most available data come from cross-sectional surveys or facility-based records, which may not capture the true burden in rural communities.

What this study adds: Conclude by clearly stating how this study fills these gaps—by leveraging HDSS data and verbal autopsy to provide robust, community-level evidence on both trends and causes of under-five mortality in Southwest Ethiopia.

**3. Clarify the period for verbal autopsy (VA) data.**

In the abstract (lines 52–54) and methods (lines 122–124), the VA period is stated as 2009–2016. However, in the results (lines 206–208) and limitations (lines 361–363), the period is given as 2008–2015. Please verify and unify these dates throughout the manuscript (including tables and figures) to ensure consistency.

**4. Address discrepancies in VA data years in figures and tables.**

Table 1 and Table 2 are labeled as "2009 to 2016," but the results text refers to "2008 to 2015." Please ensure all elements (abstract, methods, results, tables, and figures) reflect the same correct time frame.

**5. Provide the specific p-values or confidence intervals for the Mann-Kendall trend test.**

In lines 273–274, you report tau values and two-sided p-values (e.g., p=1.63e−05*p*=1.63e−05). Please also present these in a more reader-friendly format (e.g., p<0.001*p*<0.001) and ensure they are clearly labeled in the text or in a supplementary table.

**6. Correct minor typographical and grammatical errors.**

Line 217: "were (Delete were) occurred" – please revise to "occurred".Line 269: "alue" should be "tau value".Line 276: "Delete the" before "Figure 5" – please correct.Line 279: "Delete a" before "smooth" – please correct.Line 305: "was (Delate was) occurred" – please revise to "occurred".

**7. Strengthen the interpretation of the LOESS smoothed trend.**

You mention that the LOESS model "revealed a relatively smooth declining pattern" (lines 277–280). Please add a brief statement on what this smoothing adds beyond the raw data (e.g., confirming that the decline is consistent despite year-to-year fluctuations).

**8. Strengthen the discussion with clear implications for policy, research, and practice.**

The discussion should be restructured to include a dedicated section or paragraph explicitly addressing implications. Please ensure the following:

**Implications for policy**

Based on the finding recommend strengthening perinatal and neonatal care.

Given that infectious diseases remain leading causes in post-neonatal and child age groups, emphasize the need for integrated interventions: immunization, nutrition programs, water and sanitation, and malaria prevention.

Highlight the finding that over 60% of deceased children did not visit a health facility, suggesting critical barriers to access.

**Implications for research**

Note that VA data were only available for part of the surveillance period. What should you recommend?

Implication for practice????

**9.  Expand on the "undetermined" causes of death.**

In Table 2, a small proportion of deaths are listed as "undetermined" (e.g., 2.8% in neonates, 6.3% in post-neonates). Please comment in the discussion on whether this proportion is consistent with other HDSS studies and whether it could affect the interpretation of cause-specific mortality.

**10. Address the potential impact of the COVID-19 period.**

Although your study ends in 2019, consider adding a brief note in the discussion that the pre-pandemic trends observed may serve as a baseline for future studies evaluating the impact of COVID-19 on under-five mortality in Ethiopia.

**11. Verify the data availability statement.**

You state in lines 375–376: *"The data used in this study is available from the corresponding author on reasonable request."* However, in the submission form, you indicate that data are held by the Jimma University HDSS Coordination Office and require a formal data-sharing agreement. Please align the in-text statement with the journal’s requirement to specify the third-party data access contact. I recommend using the more detailed statement already provided in the submission form.

These revisions will further strengthen the clarity, consistency, and overall scientific rigor of your manuscript. We look forward to seeing your revised version.

We look forward to receiving your revised manuscript.

Kind regards,

Addis Eyeberu

Academic Editor

PLOS One

Journal Requirements:

Additional Editor Comments:

Dear authors, thank you for your revisions. Please address the following points to make your article rigorous and publishable.

1. Modify the title for clarity and consistency.

The title currently reads: *"Causes and Trends of Under-five Mortality: A Retrospective Cohort Study Using Gilgel Gibe Health and Demographic Surveillance System, 2005 to 2019, Southwest Ethiopia"*.

I suggest revising it to a more concise and standard format: *"Causes and Trends of Under-five Mortality in the Gilgel Gibe Health and Demographic Surveillance System, Southwest Ethiopia: Cohort Study"*. This places the study design at the end, which aligns with PLOS ONE style.

2. Strengthen the introduction to better articulate the problem, policy efforts, gaps, and study contribution.

The introduction should be restructured to clearly guide the reader through:

Burden of the problem: Global and national under-five mortality figures, with emphasis on Ethiopia's context and regional disparities.

Causes and consequences: Briefly highlight the leading causes of death and their impact on families, health systems, and national development.

Policy efforts and gaps: Discuss existing national and global strategies (e.g., SDGs, Child Survival Action), and acknowledge that despite these efforts, progress remains uneven. Emphasize the persistent data gaps—particularly the lack of reliable, long-term, community-level data on causes and trends.

What has not been done: Note that most available data come from cross-sectional surveys or facility-based records, which may not capture the true burden in rural communities.

What this study adds: Conclude by clearly stating how this study fills these gaps—by leveraging HDSS data and verbal autopsy to provide robust, community-level evidence on both trends and causes of under-five mortality in Southwest Ethiopia.

3. Clarify the period for verbal autopsy (VA) data.

In the abstract (lines 52–54) and methods (lines 122–124), the VA period is stated as 2009–2016. However, in the results (lines 206–208) and limitations (lines 361–363), the period is given as 2008–2015. Please verify and unify these dates throughout the manuscript (including tables and figures) to ensure consistency.

4. Address discrepancies in VA data years in figures and tables.

Table 1 and Table 2 are labeled as "2009 to 2016," but the results text refers to "2008 to 2015." Please ensure all elements (abstract, methods, results, tables, and figures) reflect the same correct time frame.

5. Provide the specific p-values or confidence intervals for the Mann-Kendall trend test.

In lines 273–274, you report tau values and two-sided p-values (e.g., p=1.63e−05p=1.63e−05). Please also present these in a more reader-friendly format (e.g., p<0.001p<0.001) and ensure they are clearly labeled in the text or in a supplementary table.

6. Correct minor typographical and grammatical errors.

Line 217: "were (Delete were) occurred" – please revise to "occurred".

Line 269: "alue" should be "tau value".

Line 276: "Delete the" before "Figure 5" – please correct.

Line 279: "Delete a" before "smooth" – please correct.

Line 305: "was (Delate was) occurred" – please revise to "occurred".

7. Strengthen the interpretation of the LOESS smoothed trend.

You mention that the LOESS model "revealed a relatively smooth declining pattern" (lines 277–280). Please add a brief statement on what this smoothing adds beyond the raw data (e.g., confirming that the decline is consistent despite year-to-year fluctuations).

8. Strengthen the discussion with clear implications for policy, research, and practice.

The discussion should be restructured to include a dedicated section or paragraph explicitly addressing implications. Please ensure the following:

Implications for policy

Based on the finding recommend strengthening perinatal and neonatal care.

Given that infectious diseases remain leading causes in post-neonatal and child age groups, emphasize the need for integrated interventions: immunization, nutrition programs, water and sanitation, and malaria prevention.

Highlight the finding that over 60% of deceased children did not visit a health facility, suggesting critical barriers to access.

Implications for research

Note that VA data were only available for part of the surveillance period. What should you recommend?

Implication for practice????

7. Expand on the "undetermined" causes of death.

In Table 2, a small proportion of deaths are listed as "undetermined" (e.g., 2.8% in neonates, 6.3% in post-neonates). Please comment in the discussion on whether this proportion is consistent with other HDSS studies and whether it could affect the interpretation of cause-specific mortality.

8. Address the potential impact of the COVID-19 period.

Although your study ends in 2019, consider adding a brief note in the discussion that the pre-pandemic trends observed may serve as a baseline for future studies evaluating the impact of COVID-19 on under-five mortality in Ethiopia.

9. Verify the data availability statement.

You state in lines 375–376: "The data used in this study is available from the corresponding author on reasonable request." However, in the submission form, you indicate that data are held by the Jimma University HDSS Coordination Office and require a formal data-sharing agreement. Please align the in-text statement with the journal’s requirement to specify the third-party data access contact. We recommend using the more detailed statement already provided in the submission form.

These revisions will further strengthen the clarity, consistency, and overall scientific rigor of your manuscript. We look forward to seeing your revised version.

Reviewers' comments:

Reviewer's Responses to Questions

**Comments to the Author**

1. If the authors have adequately addressed your comments raised in a previous round of review and you feel that this manuscript is now acceptable for publication, you may indicate that here to bypass the “Comments to the Author” section, enter your conflict of interest statement in the “Confidential to Editor” section, and submit your "Accept" recommendation.

Reviewer #1: All comments have been addressed

2. Is the manuscript technically sound, and do the data support the conclusions?

Reviewer #1: Yes

3. Has the statistical analysis been performed appropriately and rigorously? 

Reviewer #1: Yes

4. Have the authors made all data underlying the findings in their manuscript fully available?

Reviewer #1: Yes

5. Is the manuscript presented in an intelligible fashion and written in standard English?

Reviewer #1: Yes

6. Review Comments to the Author

Reviewer #1: All the raised comments seem to have been answered to my satisfaction. Thank you for this great work

7. PLOS authors have the option to publish the peer review history of their article (what does this mean?). If published, this will include your full peer review and any attached files.

Reviewer #1: No

---

## [Author Response · Author response to Decision Letter 2]

27 Apr 2026

Point by point response to the comments

Dear Editor,

Thank you for the comments and suggestions provided on our manuscript titled “Causes and Trends of Under-five Mortality: A Retrospective Cohort Study Using Gilgel Gibe Health and Demographic Surveillance System, 2005 to 2019, Southwest Ethiopia.” We have addressed your comments and suggestions.

The following are the comments and suggestions from the editor along our responses.

1. Modify the title for clarity and consistency.

The title currently reads: *"Causes and Trends of Under-five Mortality: A Retrospective Cohort Study Using Gilgel Gibe Health and Demographic Surveillance System, 2005 to 2019, Southwest Ethiopia"*.

I suggest revising it to a more concise and standard format: *"Causes and Trends of Under-five Mortality in the Gilgel Gibe Health and Demographic Surveillance System, Southwest Ethiopia: Cohort Study"*. This places the study design at the end, which aligns with PLOS ONE style.

Response: Thank you for your suggestion, and we have modified the title accordingly.

2. Strengthen the introduction to better articulate the problem, policy efforts, gaps, and study contribution.

The introduction should be restructured to clearly guide the reader through:

Burden of the problem: Global and national under-five mortality figures, with emphasis on Ethiopia's context and regional disparities.

Causes and consequences: Briefly highlight the leading causes of death and their impact on families, health systems, and national development.

Policy efforts and gaps: Discuss existing national and global strategies (e.g., SDGs, Child Survival Action), and acknowledge that despite these efforts, progress remains uneven. Emphasize the persistent data gaps—particularly the lack of reliable, long-term, community-level data on causes and trends.

What has not been done: Note that most available data come from cross-sectional surveys or facility-based records, which may not capture the true burden in rural communities.

What this study adds: Conclude by clearly stating how this study fills these gaps—by leveraging HDSS data and verbal autopsy to provide robust, community-level evidence on both trends and causes of under-five mortality in Southwest Ethiopia.

Response: Thank you very much for your detail comments and suggestions. We have tried to address your comments and incorporated your suggestions.

3. Clarify the period for verbal autopsy (VA) data.

In the abstract (lines 52–54) and methods (lines 122–124), the VA period is stated as 2009–2016. However, in the results (lines 206–208) and limitations (lines 361–363), the period is given as 2008–2015. Please verify and unify these dates throughout the manuscript (including tables and figures) to ensure consistency.

Response: Thank you for your comments and suggestions. Now it is corrected accordingly, all VA years are 2009 to 2016, including in tables and figures.

4. Address discrepancies in VA data years in figures and tables.

Table 1 and Table 2 are labeled as "2009 to 2016," but the results text refers to "2008 to 2015." Please ensure all elements (abstract, methods, results, tables, and figures) reflect the same correct time frame.

Response: Thank you for your comments. The dates in the manuscript regarding VA data are corrected to 2009 to 2016 in both texts and tables and graphs.

5. Provide the specific p-values or confidence intervals for the Mann-Kendall trend test.

In lines 273–274, you report tau values and two-sided p-values (e.g., p=1.63e−05p=1.63e−05). Please also present these in a more reader-friendly format (e.g., p<0.001p<0.001) and ensure they are clearly labelled in the text or in a supplementary table.

Response: Thank you for your comments, and they are corrected accordingly.

6. Correct minor typographical and grammatical errors.

• Line 217: "were (Delete were) occurred" – please revise to "occurred".

• Line 269: "alue" should be "tau value".

• Line 276: "Delete the" before "Figure 5" – please correct.

• Line 279: "Delete a" before "smooth" – please correct.

• Line 305: "was (Delate was) occurred" – please revise to "occurred".

Response: The comments are addressed accordingly.

7. Strengthen the interpretation of the LOESS smoothed trend.

You mention that the LOESS model "revealed a relatively smooth declining pattern" (lines 277–280). Please add a brief statement on what this smoothing adds beyond the raw data (e.g., confirming that the decline is consistent despite year-to-year fluctuations).

Response: Thank you for your comments and suggestions. The we addressed the comments and suggestions accordingly.

8. Strengthen the discussion with clear implications for policy, research, and practice.

The discussion should be restructured to include a dedicated section or paragraph explicitly addressing implications. Please ensure the following:

Implications for policy

Based on the finding recommend strengthening perinatal and neonatal care.

Given that infectious diseases remain leading causes in post-neonatal and child age groups, emphasize the need for integrated interventions: immunization, nutrition programs, water and sanitation, and malaria prevention.

Highlight the finding that over 60% of deceased children did not visit a health facility, suggesting critical barriers to access.

Implications for research

Note that VA data were only available for part of the surveillance period. What should you recommend?

Implication for practice????

Response: Thank you for the detail comments and suggestions that improve the manuscript. We have tried to address your suggestions and comments.

9. Expand on the "undetermined" causes of death.

In Table 2, a small proportion of deaths are listed as "undetermined" (e.g., 2.8% in neonates, 6.3% in post-neonates). Please comment in the discussion on whether this proportion is consistent with other HDSS studies and whether it could affect the interpretation of cause-specific mortality.

Response: Thank you for the comments and we have incorporated the comments.

10. Address the potential impact of the COVID-19 period.

Although your study ends in 2019, consider adding a brief note in the discussion that the pre-pandemic trends observed may serve as a baseline for future studies evaluating the impact of COVID-19 on under-five mortality in Ethiopia.

Response: Thank you for suggesting us such important point. We have included it accordingly.

11. Verify the data availability statement.

You state in lines 375–376: "The data used in this study is available from the corresponding author on reasonable request." However, in the submission form, you indicate that data are held by the Jimma University HDSS Coordination Office and require a formal data-sharing agreement. Please align the in-text statement with the journal’s requirement to specify the third-party data access contact. I recommend using the more detailed statement already provided in the submission form.

These revisions will further strengthen the clarity, consistency, and overall scientific rigor of your manuscript. We look forward to seeing your revised version.

Response: Thank you for your suggestion. We have updated the data availability statement based on your suggestion.

---

## [Editor Report · Decision Letter 2]

3 May 2026

Causes and Trends of Under-five Mortality in the Gilgel Gibe Health and Demographic Surveillance System, 2005 to 2019, Southwest Ethiopia: Cohort Study

PONE-D-25-49989R2

Dear Dr. Shiferaw,

We’re pleased to inform you that your manuscript has been judged scientifically suitable for publication and will be formally accepted for publication once it meets all outstanding technical requirements.

Kind regards,

Addis Eyeberu

Academic Editor

PLOS One

Additional Editor Comments (optional):

All comments and sugestions are addressed.
---

## [Editor Report · Acceptance letter]

PONE-D-25-49989R2

PLOS One

Dear Dr. Shiferaw,

I'm pleased to inform you that your manuscript has been deemed suitable for publication in PLOS One. Congratulations! Your manuscript is now being handed over to our production team.

Kind regards,

on behalf of

Dr. Addis Eyeberu

Academic Editor

PLOS One